# Understandings and Perceived Benefits of Outdoor-Based Support for People Living with Dementia

**DOI:** 10.3390/ijerph21081072

**Published:** 2024-08-15

**Authors:** Anthea Innes, Vanina Dal Bello-Haas, Equity Burke, Dylan Lu, Mason McLeod, Constance Dupuis

**Affiliations:** 1Gilbrea Centre for Studies in Aging, Faculty of Social Sciences, McMaster University, Hamilton, ON L8S 4L8, Canada; burkee3@mcmaster.ca (E.B.); lud20@mcmaster.ca (D.L.); mcleom14@mcmaster.ca (M.M.); dupuic1@mcmaster.ca (C.D.); 2School of Rehabilitation Science, Faculty of Health Sciences, McMaster University, Hamilton, ON L8S 4L8, Canada; vaninadbh@gmail.com

**Keywords:** outdoors, dementia, support, nature, care, social health, wellbeing, mental health, stigma

## Abstract

The importance of the outdoors for supporting well-being is recognized, but less is known about the role of the outdoors in supporting people living with dementia. The aim of this study was to examine three stakeholder groups’ understandings about outdoor-based support and care for people living with dementia to help understand what might be done to maximize the outdoors as a source of support for people living with dementia. Data were collected in Southern Ontario, Canada, between January and June 2023 via 1-1 interviews (n = 12); four focus groups (n = 17) with staff from organizations providing outdoor recreation or social programming; six in-person focus groups (n = 37); and 2 outdoor-based walking focus groups (n = 17) conducted with people living with dementia, care partners, and older adults. All interviews and focus groups, other than the walking focus groups where the field notes were used, were audio recorded and fully transcribed verbatim. Within and across data sets, thematic analysis was conducted. We report findings relating to the challenges of achieving full participation in outdoor-based activities; perceived physical, social, and mental health benefits of outdoor activities; stigma; and overcoming perceived risks. The importance of the outdoors for people living with dementia and their care partners is evident. There are implications for care policy and practice relating to the promotion of (social) health and well-being for people living with dementia.

## 1. Introduction

There are growing concerns about how to best support people living with dementia as the number of individuals with this condition is projected to increase, reflected in policy directives such as the Canadian National Dementia Strategy [1]. A common impact of dementia, even soon after an early diagnosis, can be the loss of physical function and mobility [2], as well as increased social isolation [3]. Social isolation is also recognized as a risk factor for poor brain health and dementia [4]. Finding ways to enable full participation in all aspects of community life is key to facilitating social inclusion and supporting those who have dementia and those who may be at risk of developing dementia. In 2020, it was estimated [5] that there were 597,300 individuals living with dementia in Canada. By 2030, we can expect this number will reach close to 1 million, and by 2050, 1,712,400 [5]; and the associated costs of providing appropriate care and support will also rise to an estimated $872 billion [6]. It is therefore imperative that care solutions are explored now that will help with the increasing number of people who will be impacted by dementia (primarily the person diagnosed and their care partners) and costs associated with dementia. 

The outdoors offers possibilities to respond to these concerns and to create opportunities for people living with dementia and their care partners to engage fully with nature through alternatives to traditional day care provision. There is emerging literature examining aspects of the outdoors for people living with dementia; for example, how to make the outdoors suitable for people living with dementia [7,8], with pilot studies of specific locations such as the woodlands [9], the impact of spending time outdoors in nature [10], the benefits of green space for people living with dementia [11], and a care concept growing in popularity known as Green Care Farms [12]. More generally, the impact of gardens on well-being and their potential benefits [13] have been discussed in gardening and horticulture [14,15,16]. There has also been a focus on what the benefits of nature and natural environments are for people living with dementia and their care partners, as a recent literature review demonstrates [17]. More broadly, how leisure opportunities might be developed to be more inclusive for people living with dementia [18] and how tourism may become ‘dementia-friendly’ [19], reflecting the public health concern to create dementia-friendly [20] or dementia-inclusive initiatives [21], are seen as innovative examples in the dementia field [22]. Given the range of activities and places that encompass outdoor-based support and care for people living with dementia, this project began by exploring the understanding of outdoor-based activities from the perspectives of organizations providing outdoor opportunities, people living with dementia, and their care partners. As such, this paper contributes to an emerging area of inquiry by positioning the array of initiatives that could fall under the banner of ‘outdoors’ and the potential benefits from different stakeholder perspectives, as well as the challenges of providing outdoor-based support and care.

## 2. Materials and Methods

Hearing the voices and experiences of people living with dementia is essential in research [23,24,25] and key to influencing policy and practice [26]. Therefore, we wished to use methods that would facilitate the engagement of people living with dementia, as well as their care partners and older adults who may not have received a diagnosis, in addition to the views of service provider stakeholders. To achieve this, this study adopted a multiple-methods approach using 1-1 interviews, focus groups, and walking focus groups—also known as ‘go-along interviews’ [27]. We wished to examine not only what people said about outdoor spaces but also their reactions to aspects of outdoor spaces during the walking focus groups. Dewing’s [28] consent process was followed. This involved seeking consent before the focus groups/walking focus groups, having already provided the information sheet about the project in advance to participants. During the focus groups (including walking focus groups), we were alert to non-verbal as well as any verbal indications when participants may not have wanted their views to be included. However, we found our participants to be highly engaged, for example, during the walking focus groups, participants chose, with the permission of the person, to take photographs of one another (including the research team) and the locations, and were fully involved at all times, and made statements such as ‘make sure you write that down’, as they wished their views to be recorded as well as wishing the researchers luck with the remainder of the research. Capacity to consent was assumed unless an assessment detailing inability to consent had been conducted by the gatekeeping organizations. However, all participants had the capacity to consent.

### 2.1. Focus Groups and 1-1 Interviews

Four focus groups (n = 17) and 1-1 virtual interviews (n = 12) with workers identified via a mapping exercise of organizations providing across three target areas in Southern Ontario providing outdoor-based support/programming/services were held between February and April 2023. One focus group was in person in the office of the organization, and three held virtually. Each focus group lasted around 90 min.

Participants living with dementia, care partners, and older adults were recruited via a number of local services and the researchers’ database of community partners who supported the research. Six focus groups (n = 37) comprised 15 people living with dementia (PLWD), 9 care partners (CP), and 13 older adults (OA). Five focus groups were in person, either on campus or in the local community organization premises, and 1 held virtually, each lasting around 60 min.

Focus groups were co-facilitated by two researchers, one as primary note-taker and the other as lead facilitator. Field notes were typed up within 48 h. All focus groups were audio recorded and transcribed.

### 2.2. Walking Focus Groups 

Two walking focus groups (n = 17) at outdoor sites with older adults (6), care partners, (5) and people living with dementia (6) took place in June 2023. The walking focus groups lasted around 2 h and 30 min, with time allocated at the end of this time for the group to enjoy refreshments provided by the research team. 

Each walking focus group was conducted differently due to the number of participants who chose to attend each of the two dates. We remained as one group during visit 1, and one participant offered to take photos with their own camera. Visit 2 had more participants, and we divided into smaller groups of 3–5 participants and 2 researchers. Each group walked at their own pace depending on their mobility and the areas that drew their attention. The group began and ended together to discuss their overall impressions and experiences. This format enabled a direct response to the needs of the participants while also engaging in collective discussions. During and immediately after each visit, researchers noted their observations and direct quotes.

### 2.3. Thematic Analysis

A thematic analysis of the data from each method was first conducted by two researchers to ensure we captured the issues arising from each part of the process. We then worked together to examine the initial codes and themes from each data source, resulting in our overall themes. The analysis process was led by AI and VDBH, with EB and MM. All authors reviewed and agreed the final themes. We drew on the approach of Miles et al. [29] as follows: Data condensing (reviewing participants’ views and experiences, coding sections of the narratives into initial themes, and eventually generating categories to group these themes into nodes);Data display (differences between the themes from people living with dementia, care partners, older people, and the three different types of organizations we include in our focus groups and 1-1 interviews);Drawing conclusions (using final nodes, conclusions can be drawn and verified with the narratives and relevant literature).

We then synthesized across all data collected following the same process to establish the over-arching themes as well as any unique perspectives from the different stakeholder groups.

## 3. Findings

### 3.1. Participant Characteristics

We collected demographic data for our participants across all methods of participation, as summarized in Table 1.

### 3.2. Thematic Findings

Common themes across all our data sets (organizational focus groups and 1-1 interviews, and in-person and walking focus groups with people living with dementia, older adults, and care partners) related to participants’ understandings of what outdoor based care or support might involve, the perceived benefits of outdoor based care and support, as well as the perceived challenges of engaging in outdoor based activities and support. The themes and sub-themes are presented below and include representative quotes.

### 3.3. Understandings of Outdoor-Based Care

Two sub-themes emerged relating to how outdoor-based care was conceptualized by participants—built environment and outdoor activities. 

#### 3.3.1. Built Environment

The built environment relates to human-made aspects of the landscape, including buildings, roads, neighborhoods, and cities. The design of these and how they can and should be adapted to meet the needs of people living with dementia were described. These structures or locations were planned and constructed with a purpose—but the purpose may not be amenable to creating opportunities for use of the outdoor-based spaces to provide support and care.

… sidewalks and roads and curbs, and how those all work together for people who are not only getting older, but also have other issues, like maybe mobility or perception issues, perhaps difficulties with balance, with hearing, with vision or use a wheeled device, or a cane, or something like an assistive device for maneuvering those, because I think that they are not good right now.(OFG1-P2)

If you look into the research on city planning and walkability and the things that you need to do to make it enjoyable to be outside, some of those things actually improve people’s health. There’s plenty of data around that in the city planning field.(FG1-05-D)

But I think in this particular case is really thinking about the external environment and opportunities within parks and outdoor spaces that you could intentionally design and support and provide care to people living, I guess, with dementia is what we’re thinking of….(OFG1-P1)

The nature and design of the built environment influenced whether the participants described a location as favorable for outdoor-based activities. 

#### 3.3.2. Outdoor Activities

Activities in an outdoor setting were understood as a formal activity promoting engagement with the outdoors, in particular, natural environments. A range of activities were described by organizations:

any kind of activity that is getting people outside, so could be outdoor walking outdoor on our patio as well as in our garden spaces(OIN-3)

…any social activity outside getting fresh air, whether it’s gardening, social behaviors court sports, football, walking, cycling, running any kind of outdoor activities…(OIN-6)

Older adults, care partners, and people living with dementia also talked about less formalized activities, such as walking, as an important activity either on their own or together in locations they enjoy:

I like walking, so I walk on my own.(FG6-32-D)

I think a lovely place probably people know about is walking along the lake, especially in the summertime. I mean, (FG6 -36-D) love ships, and there’re a lot of ships coming in.(FG6-37-CP)

Participants highlighted activities that were both formal and informally designed that take place in an outside environment, location, or setting. One participant noted people may be more likely to engage with others in natural outdoor environments: 

It’s not an institution. It’s something that is natural. It’s something that maybe people are more comfortable connecting with other people around.(OFG2-P2)

The benefits of outdoor-based locations for care and support received much discussion and agreement as to how being outdoors could benefit the person living with dementia, as we will now discuss. 

### 3.4. Perceived Benefits of Outdoor-Based Care

Participants described many positives of outdoor-based locations to experience care and activities for older adults and individuals living with dementia. Perceived benefits related to mental, social, and physical well-being. 

#### 3.4.1. Mental Well-Being 

Participants discussed several positive mental benefits being outside has on individuals living with dementia, including emotional well-being, and as a place where mood and emotions could be experienced in a positive manner. 

I think for me, being outdoors is very peaceful and serene.(FG1-07-D)

You can use your head. Do you know what I’m saying? The cobwebs are gone and you can sort of get a fresh start, you know?(FG6-30-CP)

Just breathing in that air, you know, just sitting there, even closing your eyes and, and listening to and listening to the quietness of, of nature.(OIN-2)

Participants noted that when an individual experiences and engages with outdoor spaces, their mood improves, a sense of freedom is experienced, and feelings of acceptance and connections to people and their communities are realized. 

But if we can get out, if I can take him for a walk somewhere, especially like parking a couple of blocks away when we have an appointment, and him walking, he does far better. His mood is just so much better after having walked. Whether it’s a difficult walk or raining, or anything. It just is way better, because if he’s stuck inside like today, I know that I’ll have a rough day because he will.(OFG1-P2)

You can get fresh air. You can get out of your house and see the nature. I enjoy that especially with my daughters.(FG4-18-D)

And poignantly, as a person living with dementia stated, they recognized that they may not remember tomorrow but they have enjoyed the benefits of being outdoors in the moment:

I have enjoyed today so much, but tomorrow I might not remember, I wish I could remember but I know I might not. But today has been beautiful.(Quote from WFG1-02-PLWD)

The outdoors was also seen as an opportunity to engage in activities that brought mental and physical benefits, for example:

Longer life, healthier body, better mental health. Making life worth living. You can get rid of your telly for the summer.(FG1-04-OA)

The place I walk … there’s a beautiful field and lots of trees around it. I just stand there because I’m from—raised in [another province] and it reminds me of home. So, I say a little prayer and that this is [home province] to me. Then I turn around and go back. (FG5-27-D)

Gardens and gardening received attention from all groups of participants. Specific therapeutic benefits of being in a garden were raised: 

Gardens signify calm. Just by definition, they’re places of comfort, calm, and respite.(OFG3-P5)

Walking in gardens is good for everybody.(FG1-01-D)

Inner peace. This place [name of garden] gives you inner peace.Quote (WFG1-02-PLWD)

Gardening activities were perceived as an opportunity for therapeutic benefit and to give a sense of purpose for individuals living with dementia. 

There is definitely the whole therapeutic exercises that comes with gardening, both mentally and physically. So that is something we have been involved in for a very long time which benefits seniors, the benefits, people of all walks of life.(OFG4-P1)

She always gives me stuff to do outside [Laughter] like cutting the grass.(FG4-22-D)

.. I think there’s definitely a like people have said just the reduction of stress, like just being able to do that kind of work [gardening], and really just like to stop your mind from worrying sometimes, and just calm your like blood pressure.(OIN-3)

Participants also spoke about missing their garden and ability to do gardening-based activities, for example:

I miss having a garden. I moved into a condo in [name of area]. We are on the ground floor, but we have no garden. It is all concrete and not, I can’t garden.Quote (WFG2-06-PLWD)

Participants noted the calming and therapeutic effects of being outdoors, for instance, the outdoors being a place of comfort, calming the mind, and helping reduce stress and emotional unrest. Gardens and gardening were highlighted in particular as places and activities providing individuals a space to relax and calm their mind as well as an opportunity to participate and engage in an enjoyable and meaningful activity. 

#### 3.4.2. Social Well-Being Benefits

Connecting with others, either formally through structured social activities and programs or informally, was also perceived as a positive benefit of being outdoors. Participants discussed the opportunities for individuals to engage socially with each other, either with staff or other community members, through engaging in organized outdoor activities. This social connection could be purposeful:

They put on a barbecue for all their volunteers and stuff, but I think that they could take advantage of a lot of these things like this as far as putting on a summer barbecue for all the Alzheimer’s people and their spouses or whatever, and get them all together and socialize and get some socializing, get them out getting something to eat, and makes kind of a party atmosphere right now.(FG6-34-OA)

…It offers more of a, it’s a social time when you can be outside of people. You have barbecues. You have picnics…(OIN-7B)

Outdoor-based activities offer chance social connections, such as conversations with another individual:

Yes. Him and his friend they’ve known each other since they were in elementary school. Then they—yes. So, they’ll going to start soon again with the biking. They bought themselves an electric bike. [Laughter] So, they don’t need—they’ll use the pedal between. Yes. I’m so proud that he’s still into biking. Yes.(FG5-24-CP)

I think the unobstructive way in which it offers social opportunity and the ability for people to kind of be together and… not to have that front and center in your face, I think, is really great.(OFG1-P3)

Yes, lawn bowling is fun. It’s actually fun, [Laughter] surprisingly fun.(FG3-14-D)

Being outdoors also allows for placing oneself to be with others and choosing to engage or not:

like the social dimension….can be passive, like you don’t necessarily have to be having a conversation with someone to have a social benefit from being outside in the world. But certainly, by being outside, you do open yourself up sometimes to you know—short conversations or interactions with other people.(OIN-5)

I’ve watched them play on the park, but I haven’t gone in and.. (FG6-35-D)… Yes. There were benches outside. It’s nice just to sit and watch.(FG6-33-CP)

passive participation of watching kids in the splash picnic feeding the birds. We also have strategies to use the outdoor influences to keep the focus is really important.(OFG2-P7)

Spending time in nature to see wildlife and being with pets and animals was described:

….where there’s nature, a lot of animals. I’m thinking about like the swans and the ducks and geese at [walking area].(FG1-07-D)

Walking, looking for the birds.(FG4-21-CP)

Yes, and I walk the dog…(FG4-22-D)

The opportunity to be part of, and build, community was also acknowledged as a benefit of getting outdoors:

And another thing, too, is that you get people from all walks of life from many different communities getting involved which can build community. They can tear down walls, build community, remove barriers. all those things. (OFG4-P1)

just having a place where they could come, and if they wanted to socialize or they wanted to meet neighbors, it’s an opportunity to do so.(OIN-10)

Some of the other things I think of is attending my grandkids’ events, so track and field, volleyball, soccer.(FG1-02-D)

Participants described the importance of connecting with others when outside to reduce social isolation. 

…then connecting with others. I think people meet each other in these spaces, and they make friends. So like reducing socialized isolation, and you know, helping with like loneliness. Yeah, and just feeling like a sense of purpose.(OIN-3)

Yes, and the kids, they are sometimes ahead of me and I say, “Stop! I am not that fast [Laughter] so you can stop and stay with me.” So, chit-chat with them and it’s really enjoyable.(FG4-18-D)

They [people living with dementia] often fall into isolation and isolation is very detrimental to the escalation of the disease. So by having them in, you know, communicating and being involved with other people is going to be good for brain chemistry and good for them overall. So being a part of a group being with others is healthy and it, you know, takes away that isolation that they often find themselves feeling.(0FG4-P1)

The potential for and benefits of social activities and interactions, whether planned intentionally and structured or more fluid and flexible with the choice of connecting, observing, or simply being around others, promotes an engagement with a space/place and opportunity to connect with other people. 

#### 3.4.3. Physical Well-Being Benefits

Physical benefits of being outdoors include the positive benefits when moving one’s body. The physical benefits an individual can experience were discussed in relation to exercise and mobility:

physical certainly like whether you’re walking or riding a bike, you know, even if you just like walk to the park, and then the down on a bench for a while, just like being outside free, fresh air.(OIN-5)

I just want to be out and running around, but sometimes, I run too far. [Laughter].(FG6-35-D)

I also think the physical benefits of even being outdoors, like, you’re going to have to get up and walk or move in some way.(OFG1-P3)

Similar to perceived social benefits, physical benefits can take place in structured and formal ways, such as attending a class, playing a team sport, or simply taking a leisurely walk in nature. 

I was just gonna say you could do so many things you could do, even like a photography class, You could do dance or motion.(OFG1-P1)

…outdoor boule courts, tennis courts, pickle ball… informal spaces that we provide to people in our community, but we also have some structured activities that we do host in a variety of outdoor locations.(OIN-8)

Participants described various physiological benefits an individual can experience from being in and interacting with the natural environment. 

It’s always great to be outside. I think it’s super important for everybody to get the fresh air, and the vitamin D is super important… nothing’s like the sun.(OIN-6)

Exercise. Makes you hungry makes you sleep (FG3-11-OA) … Yes, makes sleeping and stuff, it helps, yes.(FG3-13-CP)

it’s just natural light, and you’re increasing your oxygen that helps release those happy hormones…”(OIN-7C)

Physical benefits of outdoor-based activities can be mobility based, informal, or formally structured and facilitate various well-being benefits for individuals living with dementia. 

### 3.5. Perceived Challenges to Participating in Outdoor Based Activities 

While the benefits of being outdoors and involvement in outdoor pursuits were recognized, several challenges were identified that could limit the ability to facilitate and participate in outdoor-based support and care activities. The challenges identified included the natural elements, transportation/distance, cost as a financial barrier, and access barriers. 

#### 3.5.1. Natural Elements

Being outside comes along with being in an unpredictable natural environment, which can impact outdoor-based support and care, for example the weather, temperature, and seasons were described as factors that can cause inconveniences and significantly affect planning to be outdoors:

Last summer, the bocce group that I was participating with stopped because it was too hot. To get outside at 11:00 in the morning when it’s going to be 30 degrees or more, she makes me suggest that some of the infrastructure necessary for her to join the outdoors is going to be like a cooling center, [Laughter] fountains you can splash yourself in. Some recognition of the climate change is probably important.(FG1-05-D)

… if we see that its too hot… we do have policies … that we don’t go out, outdoor programing if its below a certain temperature with the wind chill and higher [than]… 28 degrees with humidex.(OIN-1)

Yes, [Laughter] and it’s not safe. The thing is, do I go out and risk falling and maybe end up in the hospital or have something wrong with me, or do I stay in? Like when we get the snowstorms, we just put the fireplace on and we’ll be here, watch TV. We’re not going anywhere. Do you know what I mean?(FG6-30-CP)

Or as one care partner put it:

Unfortunately, all the outdoor stuff is hinging upon the weather.(FG4-19-CP)

Participants discussed the perceived safety risks of the weather for people living with dementia: 

I’m gonna point out our extreme weather… extreme weather is hazardous to their health right?(OIN-7C)

I had to think of breathing, and all of the things that come and get harder as a person is older.(OFG2-P1)

there’s no trees…so you end up cooking… people talking, getting sunstroke, getting heat stroke and … sunburned.(OIN-1)

The risk of falling in icy or dry, dusty conditions was described:

You don’t really want to go there in 30-degree weather. It’s in full sun. The natural areas are great, but generally we’re at the top of a hill right now… To get into those natural areas, it’s down a hill and then back up a hill, and it can be slick.(OFG3-P5)

It’s huge. It’s not a path, it’s huge, like you can drive down it, well the staff drives down it but yes, it can be icy. We’re not going to take a chance and we can’t fall, we don’t want to fall.(FG2-08-D)

Because of the challenges and the identified risks, the need for planning ahead, having a back-up plan, and consideration towards specific locations, natural and built environmental elements when planning was described: 

Weather. What’s the weather got to have a backup. Is it too hot, too humid. Is there water in nearby, or there washrooms nearby? I mean all the venue and logistical stuff.(OFG2-P4)

So, you know we’re always like keeping an eye on the weather for the week, and like checking in with people beforehand. If it’s not looking good, we might reschedule. So there’s has to be some flexibility in the schedule to be able to cancel or move a time if necessary.(OIN-5)

Due to the unpredictability of natural elements and the perceived safety risks for people living with dementia, organizations aimed to have back-up plans for their outdoor-based activities. 

#### 3.5.2. Transportation/Distance

Participants expressed concerns with transportation with a lack of options (or expensive options) preventing people from participating in outdoor-based programming. 

I think sometimes the challenges is just their ability of transportation, of getting there. You know. That is always going to be a barrier for seniors in general, especially seniors struggling with some level of dementia. So be able to get there, be able to get home safely.(OFG4-P1)

I don’t drive anymore because I’m not safe. I am not safe driving. I had an accident once and I figured out why. I’m just not safe driving now. I can’t concentrate long enough. I’ve got my mind in, “Oh, I’m going to make that piece—I’m going to sew that when I go home. I’ve got a great idea for my next art piece.” Yes, I cannot drive now, but I’m a great bus expert, if I just remember where the hell I’m going.(FG1-04-OA)

Public transportation could be a barrier, especially in rural or smaller communities: 

…in the rural areas transportation is always the issue… we don’t have any buses, so you’re either coming by car or taxi or DARTS. So that is a struggle for a lot of people to get to our courts if they don’t have their own transportation.(OIN-6)

Well, I could tell you when I found out that when I was told by my doctor that I can’t drive anymore, that was the most devastating thing to facing what’s happening to me right now. There are alternatives but not every place. [County] have a little thing called [County] Transit, and that—and they also have a special transit for people that have wheelchairs and things like that. They have one for that too. It’s very inexpensive. So that’s what I use now when I go to the pool because the pool isn’t here in [village]. That transit, you have to book it. You can’t just call them up that day and say, “Could you come pick me up at 8:00?” That kind of thing. You book it ahead. My daughter books them all for me. [Laughter] So that works. Other than that, I don’t know what I would do. Like I said, I was so devastated that I couldn’t drive anymore because… that was my whole independence.(FG1-03-D)

Transportation as a barrier to participation is a challenge that can be addressed by finding locations that are accessible, but the lack of transport to natural beauty spots can restrict the range of outdoor locations that are accessible. 

#### 3.5.3. Financial Cost

Costs associated with attending outdoor locations (cost to the participant) included the cost of transport and knowledge about how to receive discounted or free travel, as this excerpt from a focus group discussion illustrates:

I don’t think it’s common knowledge, though. No, I certainly don’t know what the process is. (FG6-33-CP). No, I haven’t heard about it either (FG6-37-CP) [Context, about learning about free bus transportation to enable people living with dementia to get to outdoor locations] I’ve never heard of that. So where would you go?(FG6-36-D)

Costs to facilitate (cost to the organization) created financial barriers: 

…these kinds of programs are its its a challenge to continuously fund them… have to be able to continue to like resource the space… with tools and plants, making sure if things get broken like the little greenhouse or the shed … someone can fix those things… all of those pieces around maintenance and can continuance of programs.(OIN-3)

Others expressed concern for the costs incurred for older adults and people living with dementia to join outdoor activities: 

The programs that I run, as I mentioned earlier, are revenue-generating. So, any of these guided programs, there are fees associated with them.(OFG3-P5)

It’s just..…parking. So, I guess it depends on how they’re getting here, but yes, our trail system is free.(OFG3-P1)

I think there’s so much out there, but the availability and the knowledge and then there’s always if it’s an event like that, something that’s financial, the cost involved.(FG1-07-D)

Participants described the importance of eliminating or lessening the cost to participants:

We are very lucky that we’re able to operate with no charge programs both for the social Rec programs for the education programs and that’s you know that that helps allow people of all backgrounds to be involved.(OFG2-P7)

That’s what the government’s done, and I find it disgusting, really. The government, every time you would turn the TV on, the government will be telling you, especially for seniors, to get out and walk in the nature and everything else, and then every place that you want to go walking, they charge you for it …. they started charging all the way down the strip to park there.(FG6-34-OA)

the adaptive bike is free to participate. We don’t want there to be any financial barriers to access.(OIN-5)

Costs can limit an organization’s ability to offer programs and can also be barriers for participants to engage in outdoor-based care. One way organizations try to overcome the barrier of transportation is to run activities in a central location accessible by walking or public transit. 

#### 3.5.4. Physical Access Barriers

Organizations recognize and try to address limitations in the physical environment to create inclusive and accessible spaces, as discussed in relation to the built environment, however, the physical environment can continue to be a barrier to participation for some individuals. 

we’ve tried to have like wheelchair accessibility for pickleball courts… the openings of your gates where people can sit if you have accessible picnic tables… needs to be considered and worked, we’re not good at it and we’re getting there, we are getting there, but we’re still not good at it for, just to be honest with you.(OIN-6)

Our location is on a street with no sidewalk, so even walks are not a possibility, and are our parking lot not very level. So it’s not safe.(OIN-7C)

People living with dementia discussed the physical access barriers they encountered when trying to avail outdoor-based opportunities:

The other thing is, they’re [outdoor performances] usually in the early evening, and by the time they’re over… …it’s dark and I find getting [location]- because it’s not well lit, I really find it difficult. Because even just going down to [location] there’s a way you can get out, cut across the roof and go down and beat everybody, and I can’t even do that because the lighting is so poor.(FG1-04-OA)

(WFG2-17-CP) and (WFG2-16-D) commented that the paths [in a garden] with cobblestones were not “the best”—the probing cane is meant to be swept side-to-side in front of the person using; the cane gets caught in the cracks between the stones; there are similar issues when there are cracks in the cement or grates or other aspects of the pathways that are not smooth. Fieldnote WFG2.

Having easily available [e.g., not only one at the end of a 3 km walk ] access to bathrooms that are identifiable to people living with dementia is also important:

I will tell you I was at—just as on and off topic, I was at a seniors center a couple days ago. I can’t remember, but it was one day this week anyhow. They had the door for the women’s bathroom and all around that door and the door was pink, and then the men’s bathroom, all around the men’s bathroom was blue. I thought it was phenomenal.(FG1-02-D)

I do know that our washrooms aren’t accessible yet. I shouldn’t say most people. I don’t know, the majority of people with dementia, whether they require a larger space, maybe not. There is that and. The building is actually confusing.(OFG3-P3)

Being able to identify walking trails and the length of time to complete them was off putting, as this care partner highlights:

Signage, too. Yes, we’ve been on a couple of trails and it’s like, “I thought you said this was a short trail.” “Well, I don’t know. I’ve never been on it before. I’m not sure.” The sign doesn’t really tell us how much farther we’ve got.[Laughter] (FG3-13-CP)

Limitations of and within the physical space can be physical barriers to participants’ ability to access and engage in outdoor-based activities. The physical environment presents challenges that are difficult to change because of financial limitations, lack of city infrastructure, or lack of awareness of particular physical barriers to promote access to and use of outside spaces.

Additional barriers relate to physical ability and mobility associated with aging bodies:

I had both knees replaced in the last couple of years, so I tend to be a little bit more careful in terms of myself, so I don’t go out as often as perhaps I should with her, but now she doesn’t go out unless it’s supervised.(FG6-31-CP)

I know when I started not being able to walk, I like to still get my fresh fruits and vegetables on my own because I can walk down one aisle and that’s about it ….(FG1-07-D)

The stigma (including self-stigma by the person living with dementia and their care partner) attached to dementia that is hard to overcome was discussed:

(FG6-35-D) and I belonged to a golf club. Typically, we play with our old friends. We play together occasionally, but with the dementia, it’s got to the point where she can’t play with friends… She doesn’t know what to do…. You have to put the tee on. You have to put the ball on the tee. You have to aim it at the right direction in terms of prompting, and even then… Some of the ladies at the golf club have been really good with her and sort of helped her, but last year, we whittled down to one because it was just too stressful.(FG6-31-CP)

….the Alzheimer’s just kicked in and that was a mess. Then I said, “Okay. Enough, I can’t do this…..” It was kind of scary.(FG6-32-D)

… friends start to contract, right? It’s a two-way thing. They may not like to deal with somebody with dementia, because sometimes, they’ll just go, “I don’t want to see that.”(FG6-06-CP)

Physical mobility challenges and the belief that dementia makes it difficult to continue from the perspective of how others might perceive them or a self-perception of loss of ability were considerations and contributed to deciding not to access and participate in outdoor-based activities that could provide support and care.

## 4. Discussion

This research aimed to explore three stakeholder groups’ understandings about outdoor-based support and care for people living with dementia to help understand what might be done to maximize the outdoors as a source of support for people living with dementia. Our findings demonstrate commonalities across all stakeholder groups about what outdoor-based support might involve, and, although there were different ideas about the types of activities that may be enjoyable, there was an overall view that spending time in nature involved in outdoor activities that promote connections with both people and places could provide support to people living with dementia. What constitutes outdoor support and care was conceptualized in relation to the built and natural environment, as were the types of activities that can be undertaken outdoors, both structured and unstructured. The benefits of engaging in outdoor-based locations and activities were wide ranging, reflecting participants’ interests, local outdoor natural resources, and built spaces, but had in common the reported impact of benefitting mental, social, and physical health and well-being or a combination thereof. Our findings reflect the evidence base in relation to the argued benefits of being in nature [10,17], a garden, and engaging in gardening tasks [1,6,14,15], or, more broadly, farming activities and horticulture opportunities provided by care farms [12], but demonstrate the importance of outdoor-based activities that are meaningful and give purpose to the person living with dementia and their care partners. Participants talked about structured outdoor opportunities as well as the everyday self-initiated activities, such as walks and getting into their communities. The opportunities for connections from being out of the house, whether with other people, animals, or passively watching others and gaining enjoyment from, for example, watching children play or others participating in sports or simply being in nature, were perceived as beneficial in many ways by participants and provide a taken-for-granted way to address the isolation that often accompanies dementia and the associated loneliness [3]. Broadly, getting outside is a way to achieve the aspiration of dementia-inclusive communities [21] that do not necessitate a particular design and planning challenges if communities are built in ways that promote access to local green spaces, shops, and day-to-day services, such as doctors, surgeries, and pharmacies. However, the benefits described for mental health of feeling ‘calmer’, ‘serene’, and generally at peace come from entering outdoor spaces that are calm and nature focused [17]. Being around plants, wildlife, trees and flowers, and areas with natural beauty were highly valued by our participants. Where one lives will influence access to such natural areas that may bring the associated mental health benefits—and this is when transport becomes important. People living with dementia and their care partners reported being unable to drive anymore, an issue echoed by organizations who had attempted to bring people living with dementia together in nature settings that were only accessible via driving. The lack of public transit or affordable public transit to places where participants would like to spend time in nature is problematic, as are government schemes of parking charges that may deter those on limited budgets with low disposable income for leisure activities or simply to be able to park in a safe place that provides a good walking surface in a place they find enjoyable, be it lake, beach, or countryside. This demonstrates the need to consider how to finance the cost of participating in outdoor opportunities, even something as simple as a walking area that older adults and people living with dementia find safe and accessible. The limited time window of being able to safely access the outside in countries such as Canada with extreme temperatures of heat in the summer and ice and snow in the winter also necessitates making the most of the seasons and creating ways for people living with dementia, and any other groups on low income, to be able to access walking areas when seasonally safe. Charging for parking in beauty spots in the summer months may be a source of revenue for local governments but presents an initial barrier when low disposable income and seasonal safety considerations are simultaneously at play. This example alone highlights the issue of equity of access to nature and outside spaces that is compounded by age-related mobility limitations and the stigma those with dementia report facing in their encounters in outdoor groups (eg, golfing) or outdoors in their communities more generally.

Barriers to accessing services and participation in formal and informal services due to transportation, cost, and physical access barriers, as well as age-related mobility and fear of encountering stigma related to dementia, are issues well documented in the generic dementia literature about accessing services (e.g., [30,31]) that are often compounded in more remote and rural locations [32,33] or particular types of services (e.g., tourism [19] and leisure [18]). However, given the many benefits of being outdoors, it is important to consider how to promote access to such spaces to support the well-being of people living with dementia, address national dementia strategies [1], and address the associated anticipated challenges of providing appropriate care in the future [6]. The outdoors and nature create an environment that brings health benefits providing the challenges are addressed.

Participants’ concerns about the natural elements and the need to prepare for potential changes in weather or the extremes of heat or ice and snow demonstrate practical considerations often outside the control of the organizations or participants. However, they are indicative of concerns about safety, risks, and practicalities and do limit the options that may be available. Here, much can be learned from the wider discussions and reviews of the literature about risk aversion and appropriate risk-taking in relation to dementia care and the importance of including the perspectives of the person living with dementia when coming to decisions about what may or may not be perceived as safe [34] and how to best promote the agency of the person living with dementia in deciding to participate or not. What is of interest from our findings is the due diligence around being outdoors and safety that all participant groups discussed. Participants living with dementia (and their care partners) were aware of the dangers of going out in hot or icy weather and therefore able to monitor what they felt would be ‘risky’ without the intervention of the organizers of any outdoor activities. This suggests that greater attention should be given to the wishes of those with the diagnosis, as their ability to identify risk was apparent. This is an area worth further research beyond our limited sample size.

The benefits discussed by study research participants of outdoor-based support suggest that attention should be given to creating opportunities for engagement of people living with dementia. These opportunities create benefits for well-being within different domains of mental, social, and physical well-being and are therefore likely to be beneficial at an individual level. The outdoors provide a way to address the dementia-inclusive/friendly community movement, innovative ways [22] to meet the needs of people living with dementia and their care partners, and create a way to address the care challenges documented in both Canadian [1] strategy and world public health policy [20], guiding governments across the globe to respond to the support needs of people to enable them to remain at home and in their communities for longer.

This study is limited in its context, a particular region in a particular province in Canada, however, the benefits of the outdoors were explored with different stakeholder groups and are likely to resonate with organizations and people living with dementia and their care partners in other countries. Therefore, our aim of exploring the perceptions of outdoor-based support and care could provide the basis for future research and practice.

## 5. Conclusions

This research paper highlights the complexities of understandings around what outdoor support and care may entail for people living with dementia from the perspectives of both organizations providing outdoor- or dementia-related care and support opportunities as well those engaging in the opportunities—people living with dementia, care partners, and older adults. As such, it adds to the body of knowledge on ways to support community-dwelling people living with dementia. The commonalities regarding the perceived benefits of outdoor-based opportunities for people living with dementia across all participants demonstrates the value to mental, social, and physical well-being, of being outdoors, and engaging in outdoor-based activities. It also sheds light on the often taken-for-granted, features of a location that can alter experiences and can create challenges to the inclusion of people living with dementia in outdoor spaces and locations [8,18]. This research demonstrates the need for organizations’ roles in planning future access to outdoor-based opportunities for people living with dementia [21]—as well as the considerations that care partners and people living with dementia also factor into their decision-making about venturing outdoors and accessing outdoor-based activities beyond leisure-based evidence to date [18]. The range of benefits perceived by all participants echoes previous research examining particular outdoor opportunities such as gardening [11,13,14,15,16] or being in nature [9,10] and highlights that finding solutions to accessing outdoor-based places and activities would ultimately positively influence the support and care model options available for those living with dementia and enhance mental, social, and physical well-being. It is important to note that all participants, including those living with dementia, were aware of the potential risks and safety issues of being outdoors, particularly related to extreme weather temperatures, this demonstrates that people living with dementia are aware of risks and have an ability to weigh the personal benefit in relation to any risk, eg, of a fall. This awareness of risk is an area worthy of further research to ensure that the viewpoints of those with the diagnosis are heard and respected. More research is required to examine the types of outdoor provision that may bring the highest level of benefit to people living with dementia, taking into account a cost-benefit analysis.

## Figures and Tables

**Table 1 ijerph-21-01072-t001:** Participant Characteristics.

Category	Focus Groups with Organizations (OFG) (n = 17)	Organization Interviews (OIN) (n = 12)	Focus Groups with Older Adults (OA), People living with Dementia (PLWD), and Care Partners (CP) (n = 37)	Walking Focus Groups with Older Adults (OA), People Living with Dementia (PLWD), and Care Partners (CP) (n = 17)
**Gender**				
Female	12	10	25	13
Male	5	2	11	4
No response	N/A	N/A	1	0
**Age**				
21–30	0	2	0	0
31–40	3	0	0	0
41–50	6	5	0	0
51–60	4	3	2	3
61–70	4	0	12	9
71+	0	0	22	5
No response	0	2	1	0
**Highest Level of Education**				
Community College or Technical School	4	4	N/A	N/A
Bachelor’s Degree	8	5	N/A	N/A
Graduate Degree	5	1	N/A	N/A
No response	0	2	N/A	N/A
**Years of Service in Role/Organization**				
Less than 1 year	1	2	N/A	N/A
1–3 yrs.	4	3	N/A	N/A
3–5 yrs.	1	0	N/A	N/A
5–10 yrs.	4	1	N/A	N/A
10+ yrs.	7	4	N/A	N/A
No response	0	2	N/A	N/A
**Dementia**				
Yes	N/A	N/A	15	6
**Years Since Diagnosis (Dementia)**				
Less than 1 year	N/A	N/A	2	0
1–2 yrs.	N/A	N/A	2	2
2–3 yrs.	N/A	N/A	4	2
3+ yrs.	N/A	N/A	7	0
No response	N/A	N/A	0	1
**Care Partner**				
Yes	N/A	N/A	9	5
**Older Adult Only**				
Yes	N/A	N/A	13	6

## Data Availability

Anonymized data may be available upon reasonable request but would need to be confirmed via the ethics board.

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
