# Peer review of "Understandings and Perceived Benefits of Outdoor-Based Support for People Living with Dementia"

_ijerph, 2024, doi:10.3390/ijerph21081072_

Round 1

Reviewer 1 Report

Comments and Suggestions for Authors

This paper addresses a relevant but under-investigated research topic: the potential impact of outdoor activities on the well-being of people living with dementia. The authors nicely combined individual interviews, focus groups, and walking focus groups to explore this topic with a diverse group of stakeholders, including people living with dementia, care partners, older adults, and staff from organizations providing outdoor recreation or social programs.

The paper is well-written and easy to read, offering a comprehensive review of relevant literature and context for this research. The results section provides a broad understanding of the perceived benefits and barriers to outdoor activities, reporting on physical, social, and mental health benefits as well as stigma and perceived risks. The discussion section is also well-crafted, addressing how to improve access to outdoor activities for people living with dementia based on the data collected.

I would, however, like to make a few suggestions. Probably due to the qualitative nature of the analyses and the richness of the data, the results section is very long (10 pages), whereas the introduction and discussion sections are quite short (approximately one page each). It would be beneficial to rebalance the length of these sections. For example, summarizing some results in tables or supplementary data would also help make the information clearer and easier to navigate.

Among the aspects I would like to see addressed are:

  • Combining individual interviews, focus groups, and walking focus groups is an interesting approach. What motivated the researchers to use and combine these methods? Specifically regarding walking focus groups, did the authors gain specific insights from this approach?
  • Address the question of what is specific to outdoor activities versus what is generic to leisure activities. What specific insights were gained from this research?
  • Regarding participants, a table would help clarify who participated in each research activity. How was the distribution of participants decided for each research activity?
  • Results section: There appear to be some typos (e.g., p4 l155; p7 l312).

Author Response

Reviewer 1

This paper addresses a relevant but under-investigated research topic: the potential impact of outdoor activities on the well-being of people living with dementia. The authors nicely combined individual interviews, focus groups, and walking focus groups to explore this topic with a diverse group of stakeholders, including people living with dementia, care partners, older adults, and staff from organizations providing outdoor recreation or social programs.

The paper is well-written and easy to read, offering a comprehensive review of relevant literature and context for this research. The results section provides a broad understanding of the perceived benefits and barriers to outdoor activities, reporting on physical, social, and mental health benefits as well as stigma and perceived risks. The discussion section is also well-crafted, addressing how to improve access to outdoor activities for people living with dementia based on the data collected.

Thank you for the positive review of the paper

I would, however, like to make a few suggestions. Probably due to the qualitative nature of the analyses and the richness of the data, the results section is very long (10 pages), whereas the introduction and discussion sections are quite short (approximately one page each). It would be beneficial to rebalance the length of these sections. For example, summarizing some results in tables or supplementary data would also help make the information clearer and easier to navigate.

It is usual for qualitative papers to have longer findings sections in relation to the introduction and discussion and this is also in keeping with other papers using similar methods published in this journal. We have however removed some quotations as suggested by reviewer 2 throughout the findings/results section of the paper, and therefore we have partially addressed the length issue raised in relation to the findings. It is less common to summarise qualitative findings in a table and therefore we have not adopted this suggestion but maintained the conventional writing style.

Among the aspects I would like to see addressed are:

  • Combining individual interviews, focus groups, and walking focus groups is an interesting approach. What motivated the researchers to use and combine these methods? Specifically regarding walking focus groups, did the authors gain specific insights from this approach?

A sentence has been added addressing this point on pg2 line78 of the revised MS.

  • Address the question of what is specific to outdoor activities versus what is generic to leisure activities. What specific insights were gained from this research?

This was not the focus of our study, or this paper, and therefore we have not addressed this question.

  • Regarding participants, a table would help clarify who participated in each research activity. How was the distribution of participants decided for each research activity?

We have added a table summarizing participant characteristics (as suggested by reviewer 2) that also addresses this point.

  • Results section: There appear to be some typos (e.g., p4 l155; p7 l312).

Thank you for noting these – we have corrected these (marked as track changes on the revised MS)

Reviewer 2 Report

Comments and Suggestions for Authors

Thank you for the chance to review this article. The manuscript presents the findings from a qualitative study examining the views of persons living with dementia, older adults, care partners and support organisations about the use of the outdoors and nature to support the wellbeing of persons living with dementia. Several themes were established in terms of understanding what constitutes outdoor support, activities both in a structured and unstructured settings, as well as the benefits of being in nature.

General comments

The study aim needs to be more clearly articulated early in the piece (and in the abstract). What were the authors aiming to achieve by examining the various stakeholder groups understandings of outdoor-based supports for people living with dementia (what research questions drive this study)? The authors state the outcomes include positioning the various outdoor initiatives and identification of the barriers and enablers of the provision of support and care – however, without a clearly articulated aim/research questions this claim is currently a little unsubstantiated.

In general, there are some minor wording and punctuation issues (below). Overall, a straightforward qualitative study that requires minimal revision to be ready for publication.

Specific comments

1.        Abstract

·       Research aim needs to be stated.

·       3rd sentence (line 21-24): it is unclear if the number of participants for the focused group discussions were merged. Please clarify or re-write

2.        Introduction

·       line 43-44: ‘’It was estimated…..in Canada’’ (the author should add a reference to the statement).

·       Paragraph 2: The author mentioned several emerging topics on dementia and outdoor space however no mention of the mechanisms or causal pathways of nature connection, or the evidence for the benefits of nature participation, were referred to. Addition of this background would strengthen the conceptual framing for this study.

3.        Materials and methods

·       Line 85: The author should indicate where focus group held if not virtual (organisations premises or homes of respondents etc.)? (same for the other interviews)

·       Line 90: The author should avoid the use of acronym PLWD, as dementia advocates are requesting it not be used as it is considered disrespectful of personhood.

·       How was Dewing’s process consent approach implemented in practice? Some more details would be beneficial. How does this equate with the statement about a single capacity statement? And who was the capacity assessment (going to be) performed by?

·       Walking interviews, line 96-106: what were the duration of the walking interviews?

·       Which team members participated in the thematic analysis?

·       Participants characteristics, Lines 122-142: Information here can be better represented using a table. It should also be moved to the opening of the result section

·       In the process of the convergence of the data sets, were the ‘nodes’ generated into ‘themes’ and are these the finding headings? Please articulate this a little more clearly.

4.        Results – in qualitative research we refer to ‘findings’ rather than results (assuming the journal is flexible to allow this).

The findings are generally well reported and the array of quotes are interesting and insightful. There are places where some quotes could be removed and the content summarised by the authors, for improved readability, and I recommend the authors synthesise and refine the content in this way.

·       Line 155: “? REF’’  needs to be removed.

·       Line 160-172: Indicating who said what might be helpful (what stakeholder or type of respondent).

·       Using italics for the quotes might be preferable, and avoid indenting of the following text for readability.

·       Same applies to other quotes

·       The author may consider a clearer and simple way to code quotes rather than the use of e.g. OFG-1, OIN-3 and so on.

·       The author can consider reducing the quotes, while focusing on the key message pf the respondents.

5.        Discussion

When the research aim and questions are articulated more clearly at the front of the paper, it would be beneficial to discuss your findings in relation to the aim. This would give the paper a little more structure, and highlight the significance of your findings more clearly. Furthermore, it would be interesting, and beneficial, to highlight the differences in ideas and opinions (if any) between the participant groups – as this has been shown to be quite significant in other research, especially concerning risk and safety issues and the outdoors (carers being much more risk adverse than people living with dementia). The lack of voice from people living with dementia means that more risk-adverse people are having the greatest influence, which is a significant barrier to participation. Please consider this suggestion, as it would greatly strengthen your paper, and be of benefit to the dementia scholarship community.

Conclusion: if above changes are implemented as suggested, then the conclusion will need small adjustments.

Author Response

Reviewer 2

Thank you for the chance to review this article. The manuscript presents the findings from a qualitative study examining the views of persons living with dementia, older adults, care partners and support organisations about the use of the outdoors and nature to support the wellbeing of persons living with dementia. Several themes were established in terms of understanding what constitutes outdoor support, activities both in a structured and unstructured settings, as well as the benefits of being in nature.

General comments

The study aim needs to be more clearly articulated early in the piece (and in the abstract). What were the authors aiming to achieve by examining the various stakeholder groups understandings of outdoor-based supports for people living with dementia (what research questions drive this study)? The authors state the outcomes include positioning the various outdoor initiatives and identification of the barriers and enablers of the provision of support and care – however, without a clearly articulated aim/research questions this claim is currently a little unsubstantiated.

In general, there are some minor wording and punctuation issues (below). Overall, a straightforward qualitative study that requires minimal revision to be ready for publication.

Thank you for this overall summary of the review of the paper. We have addressed each comment below and the changes are marked as track changes in the document for editor and reviewer ease of review.

Specific comments

  1. Abstract
  • Research aim needs to be stated. –  this has been added - see lines 19, 20, 21, 22
  • 3rdsentence (line 21-24): it is unclear if the number of participants for the focused group discussions were merged. Please clarify or re-write the focus group numbers were not merged, there were 37 participants in the conventional focus groups and 17 participants in the walking focus groups. This has been revised for clarity.
  1. Introduction
  • line 43-44: ‘’It was estimated…..in Canada’’ (the author should add a reference to the statement). This has been added
  • Paragraph 2: The author mentioned several emerging topics on dementia and outdoor space however no mention of the mechanisms or causal pathways of nature connection, or the evidence for the benefits of nature participation, were referred to. Addition of this background would strengthen the conceptual framing for this study. We are aware of the nature based literature (and have other ongoing studies focusing on this). However we deliberately framed this research as ‘outdoors’ therefore we did not seek to explore mechanisms or causal pathways. It is however relevant to reference the benefits of nature participation and therefore we have added ‘in nature’ to the discussion of one paper already cited (pg 2 line 59) and added an additional reference to a recent review of papers discussing nature for people living with dementia (lines 63-65) to provide the suggested context relevant to our work.

  1. Materials and methods
  • Line 85: The author should indicate where focus group held if not virtual (organisations premises or homes of respondents etc.)? (same for the other interviews) Thank you. These details have been added
  • Line 90: The author should avoid the use of acronym PLWD, as dementia advocates are requesting it not be used as it is considered disrespectful of personhood. Yes, we are aware of this body of thought, however our participants were happy with the acronym, discussed when agreeing to take part in the study, and therefore we have retained this throughout.
  • How was Dewing’s process consent approach implemented in practice? Some more details would be beneficial. How does this equate with the statement about a single capacity statement? And who was the capacity assessment (going to be) performed by? Detail has been added to the discussion on process consent as requested (lines 85-95).
  • Walking interviews, line 96-106: what were the duration of the walking interviews? Detail added
  • Which team members participated in the thematic analysis? We have added this detail as requested.
  • Participants characteristics, Lines 122-142: Information here can be better represented using a table. It should also be moved to the opening of the result section. We appreciate that this is often a preference, and started with tables to then summarise the characteristics. It wasn’t our preference as we felt the narrative was easier to follow, but we have now created a table and moved it to the beginning of the results/findings section as requested. We are happy to go with whichever presentation is preferred by the journal.
  • In the process of the convergence of the data sets, were the ‘nodes’ generated into ‘themes’ and are these the finding headings? Please articulate this a little more clearly. We have added detail (lines 129-133) to clarify.

  1. Results – in qualitative research we refer to ‘findings’ rather than results (assuming the journal is flexible to allow this). We agree with the reviewers comment but had followed the template provided by the journal where the term ‘Results’ is used and this is the term that appears to be used consistently in the journal for other qualitative research (based on our check of this on the initial writing). We have however changed this in the revised MS but are happy to comply with whatever the journal editor/publisher prefers.

The findings are generally well reported and the array of quotes are interesting and insightful. There are places where some quotes could be removed and the content summarised by the authors, for improved readability, and I recommend the authors synthesise and refine the content in this way.

  • Line 155: “? REF’’  needs to be removed. Thank you – we have removed this.
  • Line 160-172: Indicating who said what might be helpful (what stakeholder or type of respondent). Thank you – we have incorporated this suggestion into the new table. Eg Older Adult (OA) Care Partner (CP).
  • Using italics for the quotes might be preferable, and avoid indenting of the following text for readability. When we checked the journal style for previously published papers indents were used. We have therefore retained this style
  • Same applies to other quotes as above
  • The author may consider a clearer and simple way to code quotes rather than the use of e.g. OFG-1, OIN-3 and so on. By indicating the abbreviation in the new table we believe the style of presentation is clear.
  • The author can consider reducing the quotes, while focusing on the key message pf the respondents. We have reviewed the quotes and deleted a few as suggested also by reviewer 1. These are marked as tracked changed deletions in the findings section of the paper.

  1. Discussion

When the research aim and questions are articulated more clearly at the front of the paper, it would be beneficial to discuss your findings in relation to the aim. This would give the paper a little more structure, and highlight the significance of your findings more clearly. Furthermore, it would be interesting, and beneficial, to highlight the differences in ideas and opinions (if any) between the participant groups – as this has been shown to be quite significant in other research, especially concerning risk and safety issues and the outdoors (carers being much more risk adverse than people living with dementia). The lack of voice from people living with dementia means that more risk-adverse people are having the greatest influence, which is a significant barrier to participation. Please consider this suggestion, as it would greatly strengthen your paper, and be of benefit to the dementia scholarship community.

We have made reference to the broader aim at the start and end of our discussion section.

Thank you for the suggestion in relation to risk and safety. We reflect on our findings in relation to risk demonstrating that participants with dementia were able to make their own decisions about risk from participating outdoors, particularly in relation to weather.

We have also ensured that we have made reference to the nature based element as suggested in relation to adding this to the introduction and added the relevant citations we included.

Conclusion: if above changes are implemented as suggested, then the conclusion will need small adjustments.

In relation to the risk point made in the discussion section – we have updated the conclusion to reflect this as an area requiring further attention.